# Persistent eczema leads to both impaired growth and food allergy: JECS birth cohort

Kiwako Yamamoto-Hanada[1,2]*, Yuichi Suzuki[3], Limin Yang[1,2], Mayako Saito-Abe[1,2], Miori Sato[1,2], Hidetoshi Mezawa[1,2], Minaho Nishizato[1,2], Noriko Kato[4], Yoshiya Ito[5], Koichi Hashimoto[3,6], Yukihiro Ohya[1,2], the Japan Environment and Children's Study (JECS) Group[¶]

1 Allergy Center, National Center for Child Health and Development, Tokyo, Japan, 2 Medical Support Center for the Japan Environment and Children's Study, National Center for Child Health and Development, Tokyo, Japan, 3 Department of Pediatrics, Fukushima Medical University School of Medicine, Fukushima, Japan, 4 Department of Early Childhood and Elementary Education, Jumonji University, Saitama, Japan, 5 Faculty of Nursing, Japanese Red Cross Hokkaido College of Nursing, Hokkaido, Japan, 6 Fukushima Regional Center for JECS, Fukushima, Japan

¶ Membership of the Japan Environment and Children's Study Group is listed in the Acknowledgments
* yamamoto-k@ncchd.go.jp

**Data Availability Statement:** Data are unsuitable for public deposition due to ethical restrictions and legal framework of Japan. It is prohibited by the Act

## Abstract

Skin inflammation leads to altered cytokine/chemokine production and causes systemic inflammation. The systemic mechanism of atopic dermatitis (AD) is recognized to affect systemic metabolism. This study aimed to examine the relationship between early-onset persistent eczema and body weight, height, and body mass index (BMI), in addition to food allergy in a birth cohort among infants. This study design was a nationwide, multicenter, prospective birth cohort study—the Japan Environment and Children's Study (JECS). Generalized linear models were fitted for z scores of weight, height, BMI, and food allergy to evaluate the relationship between eczema and these outcomes for infants at age1, 2, and 3 years. Persistent eczema was negatively associated with height at the age of 2 years (estimated coefficient, −0.127; 95% confidence interval [CI], −0.16 to −0.095) and 3 years (−0.177; 95% CI, −0.214 to −0.139). The same tendency was also observed with weight and BMI. Early disease onset at younger than 1 year and persistent eczema had the strongest association with development of food allergy at age 3 years (OR, 11.794; 95% CI, 10.721–12.975). One phenotype of eczema with early-onset and persistent disease creates a risk of both physical growth impairment and development of food allergy. Infants who present with the early-onset and persistent type of eczema should be carefully evaluated daily for impaired physical growth and development of food allergy.

## Introduction

Atopic dermatitis (AD) is characterized by chronic skin inflammation and heterogeneous disease [1]. Several studies reported several phenotypes of AD in children [2, 3]. In Japan, 7.3% young children were diagnosed as AD from a national birth cohort [4]. AD is associated with

on the Protection of Personal Information (Act No. 57 of 30 May 2003, amendment on 9 September 2015) to publicly deposit the data containing personal information. Ethical Guidelines for Medical and Health Research Involving Human Subjects enforced by the Japan Ministry of Education, Culture, Sports, Science and Technology and the Ministry of Health, Labour and Welfare also restricts the open sharing of the epidemiologic data. All inquiries about access to data should be sent to: jecs-en@nies.go.jp. The person responsible for handling enquiries sent to this e-mail address is Dr Shoji F. Nakayama, JECS Programme Office, National Institute for Environmental Studies. URL https://www.env.go.jp/chemi/ceh/en/index.html. The authors had no special access privileges to the data others would not have.

**Funding:** This study was funded by the Ministry of the Environment, Japan. The funders had no role in study design, data collection and analysis, decision to publish, or preparation of the manuscript.

**Competing interests:** The authors declare that they have no competing interests related to the contents of this study.

various comorbidities such as anxiety, depression, and attention deficit hyperactivity disorder [5, 6]. Skin inflammation leads to altered cytokine/chemokine production and causes systemic inflammation [7]. Thereby, the systemic mechanism of AD is recognized to affect systemic metabolism. Nomura et al. [8] reported that infants hospitalized with severe AD had impaired mental and physical growth, protein loss through skin inflammation, and elevated serum interleukin (IL), including IL5, IL6, and IL12. Furthermore, early-onset and/or persistent AD is a known risk factor for food allergy based on a birth cohort study [3, 9]. We hypothesized that early-onset and persistent AD in infants may lead not only to impaired physical growth but food allergy as well because of long-term skin inflammation. This study aimed to examine the relationship between early-onset persistent AD and body weight, height, and body mass index (BMI), in addition to food allergy in a birth cohort.

## Materials and methods

This study design was a nationwide, multicenter, prospective birth cohort study—the Japan Environment and Children's Study (JECS), funded by the Ministry of the Environment, Japan [10–12]. The JECS enrolled a general population of 103,060 pregnant women in 15 Study Areas covering a wide region across Japan from the north (Hokkaido) to south (Okinawa) from January 2011 to March 2014. Eligibility criteria were as follows: 1) currently pregnant; 2) living in the Study Area for the foreseeable future; 3) expected delivery between August 1, 2011, and mid-2014; and 4) ability to understand the Japanese language. In total, 104,062 fetuses were enrolled in the JECS. The registry of the JECS is the University Hospital Medical Information Network (UMIN Clinical Trials Registry 000030786). The JECS protocols for the main study and the sub-cohort study are described on the websites of the Ministry of the Environment, Japan [13, 14]. The JECS protocol was reviewed and approved by the Ministry of Environment's Institutional Review Board for Epidemiologic Studies (#100910001) and by the ethics committees of all participating institutions (#2019–070). Written informed consent was obtained from all participants. The JECS was conducted in accordance with the principles laid out in the Helsinki Declaration and other national regulations and guidelines.

### Questionnaire

Written questionnaires were provided to caregivers during pregnancy for child participants at age 6 months and 1, 1.5, 2, 2.5, and 3 years. Caregivers answered questions regarding the child and the family.

### Outcomes

Information on each child's background and lifestyle was assessed using questionnaires in Japanese. Eczema history and Caregiver-reported physician diagnoses food allergy were obtained from questionnaires at ages 1, 2, and 3 years.

This study extracted the children's weight and height data from surveys conducted at age 1, 2, and 3 years. The LMS (lambda-mu-sigma) statistical method was used to calculate z scores for weight, height, and BMI (weight/height2) [15]. Age- and sex-specific values of L, M, and S were obtained from the Japanese growth curve criteria [16, 17].

### Statistical analyses

After excluding preterm birth, twin birth, neonatal complications, and chronic disease other than eczema and food allergy, 59,847 mother–child pairs remained for analysis (S1 Fig). A

fixed data set (jecs-ta-201901930-qsn, released in October 2019) was used for this study. Generalized linear models were fitted for z scores of weight, height, BMI, and food allergy. An identity link function was used to model continuous outcomes (z scores of weight, height, BMI), and a logit link was used for modeling the binary outcome (food allergy). The coefficients in the models provided measures for the strength of associations (compared with the reference group). Three models were fitted for each outcome (z scores of weight, height, BMI, and food allergy) for children at ages 1, 2, and 3 years. For the models evaluating the relationship between eczema and outcomes for children at age 2 years, the exposure variable eczema was classified into four groups: 1) no eczema at 1 and 2 years; 2) eczema at 1–2 years; 3) eczema only at 1 year; and 4) eczema only at 2 years. The group that had no eczema at age 1 and 2 years was designated as reference group. Similarly, on assessment at age 3 years, children with eczema during 1–3 years had eight patterns based on whether they had eczemaat age 1, 2, and 3 years or not. The status of no eczema at 1, 2, and 3 years was designated as reference group in the models. Eczema and food allergy are high multicollinearity so we did not input food allergy in the models. An assumption was made that data were missing at random. Missing data for independent variables were imputed using multiple imputation (MI) analysis with a chained equations (MICE) algorithm. The variables used for MI process included sex, siblings, maternal history of AD, paternal history of AD, and maternal highest level of education. To obtain pooled coefficients of models, 20 data sets with missing data were generated. Bonferroni correction was applied for correcting multiple testing, and the thresholds were set at 0.05/44 (0.001). For the sensitivity analysis, the same models were refitted using complete dataset.

All the analyses were performed using R software (version 4.0.3, Institute for Statistics and Mathematics, Vienna, Austria; www.r-project.org). The R packages "MICE" was used for the MI process.

## Results

Table 1 shows the baseline characteristics. Maternal history of AD was found for 15.8% of participants. Income: <4,000,000 yen/annual income was reported from 38.8 participants.

Table 2 presents the numbers of participants with AD and food allergy. At the age of 1 year, 19% infants had AD and food allergy. At the age of 3 years, 27.7% infants had persistent AD and food allergy. Most cases of AD at the age of 1 year were transient.

Table 3 presents the eczema associations with z scores of body weight, height, and BMI at ages 1, 2, and 3 years. Early disease onset at younger than 1 year and persistent eczema were evaluated for association with the child's status at ages 2 and 3 years as follows. Body weight was negatively associated with persistent eczema at the age of 2 years (estimated coefficient, −0.146; 95% confidence interval [CI], −0.174 to −0.117) and 3 years (−0.148; 95% CI, −0.181 to −0.114). Height was negatively associated with persistent eczema at the age of 2 years (estimated coefficient, −0.127; 95% CI, −0.16 to −0.095) and 3 years (−0.177; 95% CI, −0.0214 to −0.139). Also, BMI was negatively associated with height at the age 2 years (estimated coefficient, −0.081; 95% CI, −0.113 to −0.05) and 3 years (−0.058; 95% CI, −0.094 to −0.022).

Table 4 shows the associations of eczema with food allergy. Early disease onset at younger than 1 year and persistent eczema had the strongest association with development of food allergy at age 2 years (odds ratio [OR], 9.861; 95% CI, 9.115–10.668) and 3 years (OR, 11.794; 95% CI, 10.721–12.975). Late onset of eczema (diagnosis at three years of age) was less associated with food allergy development (OR, 2.373; 95%CI, 2.02–2.789) compared to the early-onset and persistent eczema.

**Table 1. Baseline characteristics for participants.**

| Participant | n | N | % |
|---|---|---|---|
| Place at recruitment | | | |
| Hokkaido | 4570 | 59847 | 7.6 |
| Miyagi | 5196 | 59847 | 8.7 |
| Fukushima | 7929 | 59847 | 13.2 |
| Chiba | 3110 | 59847 | 5.2 |
| Kanagawa | 3916 | 59847 | 6.5 |
| Koshin | 4245 | 59847 | 7.1 |
| Toyama | 3377 | 59847 | 5.6 |
| Aichi | 3343 | 59847 | 5.6 |
| Kyoto | 2512 | 59847 | 4.2 |
| Osaka | 4791 | 59847 | 8 |
| Hyogo | 3147 | 59847 | 5.3 |
| Tottori | 1877 | 59847 | 3.1 |
| Kochi | 4075 | 59847 | 6.8 |
| Fukuoka | 4424 | 59847 | 7.4 |
| South Kyushu/Okinawa | 3335 | 59847 | 5.6 |
| Mother | | | |
| Age <35 years | 44473 | 59577 | 74.6 |
| Age > = 35 years | 15104 | 59577 | 25.4 |
| Education: Middle school and high school | 20144 | 59319 | 34 |
| Education: Technical, College, University and Graduate | 39175 | 59319 | 66 |
| Income: <4,000,000 yen/annual income | 21640 | 55768 | 38.8 |
| Income: > = 4,000,000 yen/annual income | 34128 | 55768 | 61.2 |
| Health: Maternal atopic dermatitis history (+) | 9400 | 59580 | 15.8 |
| Health: Maternal food allergy history (+) | 2793 | 59580 | 4.7 |
| Child | | | |
| Sibling | 33766 | 59580 | 56.7 |
| Boys | 30051 | 59847 | 50.2 |
| Girls | 29796 | 59847 | 49.8 |

n, yes, N, variables without missing data.

## Discussion

Based on these data from a large-scale, national, birth cohort study in Japan, early-onset and persistent eczema negatively affected physical growth and created a risk of low body weight, short height, low BMI, and development of food allergy. To the best of our knowledge, this is the first report on the relationship between infant eczema and physical growth among the Japanese general population. We demonstrated that early-onset and persistent eczema phenotype was the strongest risk factor for both physical growth retardation and food allergy. This is also the first report regarding the mechanism by which eczema phenotypes are linked to body weight, height, and BMI. A systematic review and meta-analysis [18] of AD and weight status in children observed that AD overall was associated with overweight (random effects OR, 1.24; 95% CI, 1.08–1.43), obesity (random effects OR, 1.44; 95% CI, 1.12–1.86), or overweight/obesity (random effects OR, 1.32; 95% CI, 1.15–1.51). This systematic review included all phenotypes of AD, which is a heterogeneous disease with several phenotypes [2, 3]. The associations of comorbidity, such as allergic diseases—food allergy, asthma, and immunoglobulin E

**Table 2. Number of participants with AD and FA.**

| Child (age in years) | FA (−) | | FA (+) | |
|---|---|---|---|---|
| | n | (%) | n | (%) |
| **1 year of age** | | | | |
| eczema1Y (−) | 46065 | 96.2 | 1809 | 3.8 |
| eczema1Y(+) | 8861 | 81 | 2082 | 19 |
| Missing | 246 | – | 15 | – |
| ALL | 55172 | 93.4 | 3906 | 6.6 |
| **2 years of age** | | | | |
| eczema1Y(−) and eczema2Y(−) | 41592 | 96.5 | 1518 | 3.5 |
| eczema1Y(+) and eczema2Y(+) | 3832 | 72.6 | 1448 | 27.4 |
| eczema1Y(−) and eczema2Y(+) | 4093 | 89.8 | 465 | 10.2 |
| eczema1Y(+) and eczema2Y(−) | 4860 | 87 | 729 | 13 |
| Missing | 1203 | – | 107 | – |
| ALL | 55580 | 92.9 | 4267 | 7.1 |
| **3 years of age** | | | | |
| eczema1Y(−) and eczema2Y(−) and eczema3Y(−) | 39143 | 97.2 | 1135 | 2.8 |
| eczema1Y(+) and eczema2Y(+) and eczemaY(+) | 2416 | 72.8 | 904 | 27.2 |
| eczema1Y(+) and eczema2Y(+) and eczema3Y(−) | 1504 | 80.7 | 360 | 19.3 |
| eczema1Y(+) and eczema2Y(−) and eczema3Y(+) | 834 | 82.5 | 177 | 17.5 |
| eczema1Y(+) and eczema2Y(−) and eczema3Y(−) | 4049 | 89.5 | 477 | 10.5 |
| eczema1Y(−) and eczema2Y(+) and eczema3Y(+) | 1592 | 87.9 | 220 | 12.1 |
| eczema1Y(−) and eczema 2Y(+) and eczema3Y(−) | 2509 | 93.2 | 182 | 6.8 |
| eczema1Y(−) and eczema2Y(−) and eczema3Y(+) | 2481 | 93.2 | 181 | 6.8 |
| Missing | 1523 | – | 160 | – |
| ALL | 56051 | 93.7 | 3796 | 6.3 |

ALL, participants including; FA, food allergy; 1Y, age 1 year; 2Y, age 2 years; 3Y, age 3 years.

sensitization—differed among AD phenotypes. Adult populations with severe AD may gain weight after treatment with dupilumab [19]. Early-onset AD tended to be more severe disease [20]. A systematic review of 66 studies concluded that AD appeared to precede the development of food allergy. Therefore, we considered that AD occurs before food allergy. It is also known that AD can be a risk factor for developing IgE sensitization to allergens. We considered that eczema occurred first, sensitization second, and food allergy third based on the systematic review. The present study speculates that severe and persistent eczema may lead to persistent skin inflammation, producing various cytokines/chemokines from the skin that may in turn affect systemic metabolism. A previous study reported that severe childhood AD led to hypoprotenemia, hyperkaremia, and hyponatremia as leaking through the skin [8]. Homeostasis of the body could be damaged by skin inflammation. Early-onset and persistent eczema may create a risk of not only "allergic march" but also slowed physical growth. Multiple comorbidities related to eczema should be considered for this eczema phenotype [21].

In general, epidemiologic studies have limitations. Reporting biases inevitably arise. Outcome assessments were not made directly by clinicians but through a questionnaire given to caregivers. The prevalence and incidence of the disease may have been overestimated or underestimated. However, we could apply physician-diagnosis outcomes. Various past studies have used same definitions. Second, information regarding medical interventions, including medications, was not obtained, thus it was not possible to evaluate how medical interventions

**Table 3. Caregiver-reported physician diagnoses of atopic dermatitis and physical growth.**

| Age (years) | Outcomes (z scores) | | Eczema 1 year | Eczema 2 years | Eczema 3 years | Coefficient[a] | SE | 95% CI Lower | 95% CI Upper | p value[b] |
|---|---|---|---|---|---|---|---|---|---|---|
| 1 | Weight | Eczema1Y (−) | − | | | 1 | | | | |
| | | **Eczema1Y(+)** | + | | | **−0.093** | **0.011** | **−0.114** | **−0.072** | **<0.0001** |
| 2 | Weight | Eczema 1Y(−) Eczema 2Y(−) | − | − | | 1 | | | | |
| | | **Eczema1Y(+) Eczema2Y(+)** | + | + | | **−0.146** | **0.015** | **−0.174** | **−0.117** | **<0.0001** |
| | | Eczema 1Y(−) Eczema 2Y(+) | − | + | | 0.012 | 0.016 | −0.018 | 0.042 | 0.4404 |
| | | **Eczema1Y(+) Eczema2Y(−)** | + | − | | **−0.079** | **0.014** | **−0.107** | **−0.051** | **<0.0001** |
| 3 | Weight | Eczema 1Y(−) Eczema 2Y(−) Eczema 3Y(−) | − | − | − | 1 | | | | |
| | | **Eczema1Y(+) Eczema2Y(+) Eczema3Y(+)** | + | + | + | **−0.148** | **0.017** | **−0.181** | **−0.114** | **<0.0001** |
| | | Eczema1Y(+) Eczema 2Y(+) Eczema3Y(−) | + | + | − | −0.067 | 0.022 | −0.111 | −0.023 | 0.0029 |
| | | Eczema1Y(+) Eczema 2Y(−) Eczema 3Y(+) | + | − | + | −0.019 | 0.03 | −0.079 | 0.04 | 0.5230 |
| | | **Eczema1Y(+) Eczema2Y(−) Eczema3Y(−)** | + | − | − | **−0.058** | **0.015** | **−0.087** | **−0.029** | **0.000108** |
| | | Eczema 1Y(−) Eczema 2Y(+) Eczema 3Y(+) | − | + | + | −0.037 | 0.023 | −0.082 | 0.008 | 0.1068 |
| | | Eczema 1Y(−) Eczema 2Y(+) Eczema 3Y(−) | − | + | | −0.011 | 0.019 | −0.048 | 0.026 | 0.5487 |
| | | Eczema 1Y(−) Eczema 2Y(−) Eczema 3Y(+) | − | − | + | −0.005 | 0.019 | −0.043 | 0.032 | 0.7718 |
| 1 | Height | Eczema1Y (−) | − | | | 1 | | | | |
| | | **Eczema1Y(+)** | + | | | **−0.047** | **0.011** | **−0.068** | **−0.025** | **<0.0001** |
| 2 | Height | Eczema 1Y(−) Eczema 2Y(−) | − | − | | 1 | | | | |
| | | **Eczema1Y(+) Eczema2Y(+)** | + | + | | **−0.127** | **0.017** | **−0.16** | **−0.095** | **<0.0001** |
| | | Eczema 1Y(−) Eczema 2Y(+) | − | + | | −0.029 | 0.018 | −0.064 | 0.005 | 0.0982591 |
| | | Eczema1Y(+) Eczema 2Y(−) | + | − | | −0.042 | 0.016 | −0.074 | −0.011 | 0.0088 |
| 3 | Height | Eczema 1Y(−) Eczema 2Y(−) Eczema 3Y(−) | − | − | − | 1 | | | | |
| | | **Eczema1Y(+) Eczema2Y(+) Eczema3Y(+)** | + | + | + | **−0.177** | **0.019** | **−0.214** | **−0.139** | **<0.0001** |
| | | Eczema1Y(+) Eczema 2Y(+) Eczema3Y(−) | + | + | − | −0.065 | 0.025 | −0.114 | −0.016 | 0.0098 |
| | | Eczema1Y(+) Eczema 2Y(−) Eczema 3Y(+) | + | − | + | −0.098 | 0.034 | −0.165 | −0.032 | 0.0038 |
| | | Eczema1Y(+) Eczema2Y(−) Eczema3Y(−) | + | − | − | −0.055 | 0.017 | −0.088 | −0.022 | 0.0010 |
| | | **Eczema 1Y(−) Eczema 2Y(+) Eczema 3Y(+)** | − | + | + | **−0.095** | **0.025** | **−0.145** | **−0.045** | **0.0002** |
| | | Eczema 1Y(−) Eczema 2Y(+) AD3Y(−) | − | + | | −0.027 | 0.021 | −0.068 | 0.014 | 0.2007 |
| | | Eczema 1Y(−) Eczema 2Y(−) Eczema 3Y(+) | − | − | + | −0.048 | 0.021 | −0.09 | −0.006 | 0.0246 |
| 1 | BMI | Eczema1Y (−) | − | | | 1 | | | | |
| | | **Eczema1Y(+)** | + | | | **−0.069** | **0.011** | **−0.09** | **−0.048** | **<0.0001** |
| 2 | BMI | Eczema 1Y(−) Eczema 2Y(−) | − | − | | 1 | | | | |
| | | **Eczema1Y(+) Eczema2Y(+)** | + | + | | **−0.081** | **0.016** | **−0.113** | **−0.05** | **<0.0001** |
| | | Eczema 1Y(−) Eczema 2Y(+) | − | + | | 0.049 | 0.017 | 0.015 | 0.082 | 0.0046 |
| | | **Eczema1Y(+) Eczema2Y(−)** | + | − | | **−0.064** | **0.016** | **−0.094** | **−0.033** | **<0.0001** |
| 3 | BMI | Eczema 1Y(−) Eczema 2Y(−) Eczema 3Y(−) | − | − | − | 1 | | | | |
| | | **Eczema1Y(+) Eczema2Y(+) Eczema3Y(+)** | + | + | + | **−0.058** | **0.018** | **−0.094** | **−0.022** | **0.001483** |
| | | Eczema1Y(+) Eczema 2Y(+) Eczema3Y(−) | + | + | − | −0.051 | 0.024 | −0.098 | −0.003 | 0.0367 |
| | | Eczema1Y(+) Eczema 2Y(−) Eczema 3Y(+) | + | − | + | 0.053 | 0.033 | −0.011 | 0.117 | 0.1064 |
| | | Eczema1Y(+) Eczema2Y(−) Eczema3Y(−) | + | − | − | −0.041 | 0.016 | −0.072 | −0.009 | 0.0118 |
| | | Eczema 1Y(−) Eczema 2Y(+) Eczema 3Y(+) | − | + | + | 0.041 | 0.025 | −0.008 | 0.089 | 0.1011 |
| | | Eczema 1Y(−) Eczema 2Y(+) Eczema 3Y(−) | − | + | | 0.007 | 0.02 | −0.033 | 0.047 | 0.7271 |
| | | Eczema 1Y(−) Eczema 2Y(−) Eczema 3Y(+) | − | − | + | 0.046 | 0.02 | 0.006 | 0.086 | 0.0253 |

AD, atopic dermatitis; CI, confidence interval; 1 Y, age 1 year; 2 Y, age 2 years; 3 Y, age 3 years.

[a]Generalized linear models with identity link function.

[b]Bonferroni correction was applied for correcting multiple testing, and the thresholds were set at 0.05/44 (0.001).

**Table 4. Caregiver-reported physician diagnoses of atopic dermatitis and food allergy.**

| Age (years) | Outcomes | | Odds ratio | 95% CI | | p value[b] |
|---|---|---|---|---|---|---|
| | | | | Lower | Upper | |
| 1 | Food allergy | Eczema1Y (−) | 1 | | | |
| | | **Eczema1Y(+)** | **5.943** | **5.558** | **6.354** | **<0.0001** |
| 2 | Food allergy | Eczema 1Y(−) Eczema 2Y(−) | **1** | | | |
| | | **Eczema1Y(+) Eczema2Y(+)** | **9.861** | **9.115** | **10.668** | **<0.0001** |
| | | Eczema 1Y(−) Eczema 2Y(+) | **3.012** | **2.703** | **3.356** | **<0.0001** |
| | | **Eczema1Y(+) Eczema2Y(−)** | **3.962** | **3.61** | **4.348** | **<0.0001** |
| 3 | Food allergy | Eczema 1Y(−) Eczema 2Y(−) Eczema 3Y(−) | **1** | | | |
| | | **Eczema1Y(+) Eczema2Y(+) Eczema3Y(+)** | **11.794** | **10.721** | **12.975** | **<0.0001** |
| | | Eczema1Y(+) Eczema 2Y(+) Eczema3Y(−) | **7.603** | **6.687** | **8.646** | **<0.0001** |
| | | Eczema1Y(+) Eczema 2Y(−) Eczema 3Y(+) | **6.767** | **5.698** | **8.037** | **<0.0001** |
| | | Eczema1Y(+) Eczema2Y(−) Eczema3Y(−) | **3.791** | **3.392** | **4.236** | **<0.0001** |
| | | **Eczema 1Y(−) Eczema 2Y(+) AD3Y(+)** | **4.428** | **3.797** | **5.163** | **<0.0001** |
| | | Eczema 1Y(−) Eczema 2Y(+) AD3Y(−) | **2.366** | **2.013** | **2.782** | **<0.0001** |
| | | Eczema 1Y(−) Eczema 2Y(−) Eczema 3Y(+) | **2.373** | **2.02** | **2.789** | **<0.0001** |

CI, confidence interval; 1 Y, age 1 year; 2 Y, age 2 years; 3 Y, age 3 years.

[a]Generalized linear models with Logit link function (logistic regression model).

[b]Bonferroni correction was applied for correcting multiple testing, and the thresholds were set at 0.05/44 (0.001).

and disease activities may affect physical growth and food allergy. In terms of for the elimination status, we did not evaluate the details of the elimination diet. However, we believe that the Japanese guidelines on food allergy recommend minimum causal food elimination; thus, we considered that most children underwent only minimum food elimination that did not affect physical growth.

## Conclusions

This study highlighted that one phenotype of eczema with early-onset and persistent disease creates a risk of both physical growth impairment and development of food allergy. Infants who present with the early-onset and persistent type of eczema should be carefully evaluated daily for impaired physical growth and development of food allergy.

## Supporting information

**S1 Fig. Flow chart of the study participants.**
(DOCX)

## Acknowledgments

We would like to thank the children and their families for participating in the JECS. We also thank Enago, Crimson Interactive Pvt. Ltd.(www.egano.jp) for editing a draft of this manuscript. The JECS protocol was reviewed and approved by the Ministry of Environment's Institutional Review Board for Epidemiologic Studies (#100910001) and by the ethics committees of all participating institutions (#2019–070). Written informed consent was obtained from all participants. The JECS was conducted in accordance with the principles laid out in the Helsinki Declaration and other national regulations and guidelines.

Members of the JECS Group as of 2021: Michihiro Kamijima (principal investigator, Nagoya City University, Nagoya, Japan), Shin Yamazaki (National Institute for Environmental Studies, Tsukuba, Japan), Yukihiro Ohya (National Center for Child Health and Development, Tokyo, Japan), Reiko Kishi (Hokkaido University, Sapporo, Japan), Nobuo Yaegashi (Tohoku University, Sendai, Japan), Koichi Hashimoto (Fukushima Medical University, Fukushima, Japan), Chisato Mori (Chiba University, Chiba, Japan), Shuichi Ito (Yokohama City University, Yokohama, Japan), Zentaro Yamagata (University of Yamanashi, Chuo, Japan), Hidekuni Inadera (University of Toyama, Toyama, Japan), Michihiro Kamijima (Nagoya City University, Nagoya, Japan), Takeo Nakayama (Kyoto University, Kyoto, Japan), Hiroyasu Iso (Osaka University, Suita, Japan), Masayuki Shima (Hyogo College of Medicine, Nishinomiya, Japan), Youichi Kurozawa (Tottori University, Yonago, Japan), Narufumi Suganuma (Kochi University, Nankoku, Japan), Koichi Kusuhara (University of Occupational and Environmental Health, Kitakyushu, Japan), and Takahiko Katoh (Kumamoto University, Kumamoto, Japan).

## Author Contributions

**Conceptualization:** Kiwako Yamamoto-Hanada, Limin Yang, Yukihiro Ohya.

**Data curation:** Limin Yang.

**Formal analysis:** Limin Yang.

**Funding acquisition:** Yukihiro Ohya.

**Investigation:** Yuichi Suzuki, Koichi Hashimoto, Yukihiro Ohya.

**Methodology:** Kiwako Yamamoto-Hanada, Yuichi Suzuki, Limin Yang, Mayako Saito-Abe, Miori Sato, Hidetoshi Mezawa, Noriko Kato, Yoshiya Ito, Koichi Hashimoto, Yukihiro Ohya.

**Project administration:** Kiwako Yamamoto-Hanada, Yuichi Suzuki, Miori Sato, Minaho Nishizato, Koichi Hashimoto.

**Resources:** Noriko Kato.

**Software:** Noriko Kato, Yoshiya Ito.

**Supervision:** Kiwako Yamamoto-Hanada, Koichi Hashimoto, Yukihiro Ohya.

**Writing – original draft:** Kiwako Yamamoto-Hanada.

**Writing – review & editing:** Kiwako Yamamoto-Hanada, Yuichi Suzuki, Limin Yang, Mayako Saito-Abe, Miori Sato, Hidetoshi Mezawa, Minaho Nishizato, Noriko Kato, Yoshiya Ito, Koichi Hashimoto, Yukihiro Ohya.

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
