## [Decision Letter · Decision Letter 0]

21 Oct 2021

PONE-D-21-29612Persistent atopic dermatitis leads to both impaired growth and food allergy: JECS Birth CohortPLOS ONE

Dear Dr. Yamamoto-Hanada,

Thank you for submitting your manuscript to PLOS ONE. After careful consideration, we feel that it has merit but does not fully meet PLOS ONE’s publication criteria as it currently stands. Therefore, we invite you to submit a revised version of the manuscript that addresses the points raised during the review process.

ACADEMIC EDITOR:Please respond to the reviewers comment carefully to skip any further revision requirements.

We look forward to receiving your revised manuscript.

Kind regards,

Kazumichi Fujioka

Academic Editor

PLOS ONE

Journal Requirements:

3. You indicated that you had ethical approval for your study. In your Methods section, please ensure you have also stated whether you obtained consent from parents or guardians of the minors included in the study or whether the research ethics committee or IRB specifically waived the need for their consent.

4. Please include additional information regarding the survey or questionnaire used in the study and ensure that you have provided sufficient details that others could replicate the analyses. For instance, if you developed a questionnaire as part of this study and it is not under a copyright more restrictive than CC-BY, please include a copy, in both the original language and English, as Supporting Information.

6. Thank you for stating the following financial disclosure: 

"This study was funded by the Ministry of the Environment, Japan."

7. Thank you for stating the following in the Acknowledgments Section of your manuscript: 

"This study was funded by the Ministry of the Environment, Japan. "

"This study was funded by the Ministry of the Environment, Japan."

8. One of the noted authors is a group or consortium Japan Environment and Children’s Study (JECS) Group. In addition to naming the author group, please list the individual authors and affiliations within this group in the acknowledgments section of your manuscript. Please also indicate clearly a lead author for this group along with a contact email address.

9. Your ethics statement should only appear in the Methods section of your manuscript. If your ethics statement is written in any section besides the Methods, please delete it from any other section. 

Reviewers' comments:

Reviewer's Responses to Questions

**Comments to the Author**

1. Is the manuscript technically sound, and do the data support the conclusions?

Reviewer #1: Yes

Reviewer #2: Yes

2. Has the statistical analysis been performed appropriately and rigorously? 

Reviewer #1: Yes

Reviewer #2: Yes

3. Have the authors made all data underlying the findings in their manuscript fully available?

Reviewer #1: Yes

Reviewer #2: No

4. Is the manuscript presented in an intelligible fashion and written in standard English?

Reviewer #1: Yes

Reviewer #2: Yes

5. Review Comments to the Author

Reviewer #1: General Comments:

This paper used a large prospective cohort study to investigate atopic dermatitis and food allergy and body size. The weakness of this paper is that the diagnosis of food allergy is not clear. It is likely that most children did not receive appropriate oral food challenge tests during the follow-up period because the number of children with food allergies changed little between the ages of one, two, and three years. Therefore, I frankly felt that children with persistent atopic dermatitis are more likely to be misdiagnosed with food allergy. In the future, studies with a definitive diagnosis of food allergy will be more valuable than ones with large numbers of subjects. On the other hand, as far as I know, there are no studies of this size or larger, so this study is of great value. Hence, I believe that this study is worthy of publication.

Specific recommendations for revision minor:

1. Reference 18 shows that patients with obesity are more likely to have complications of atopic dermatitis. This does not seem to be relevant to the discussion of this study. For example, I propose to discuss reference "Zhang A, et al. Association of atopic dermatitis with being overweight and obese: a systematic review and metaanalysis. J Am Acad Dermatol. 2015; 72: 606-16.e4." as a foundation. Excluding severe cases, such as those involving sleep disorders, atopic dermatitis was not thought to affect the physique. It is interesting that your study found that growth retardation was present at age 3 years.

Reviewer #2: The strength of this study are impressive numbers. It’s also interesting and definite.

We know that chronic conditions could impair the growth and development of children, it is anything new here. What’s interesting, as authors have mentioned in the discussion – in other studies no such impairment was observed in AD – it will be highly welcomed to discuss this in more detailed way - why the differences between current study and previous reports are present – whether it is the age of children, comorbidities or other factors.

The coexistence of atopic dermatitis and food allergy – we don’t know what comes first in infants. The most possible course is that food allergy if the first phenomenon with the presentation of symptoms from the skin – that’s why many infants improve on elimination diet. Another aspect not discussed in the manuscript is how elimination diet could impair the growth.

For better understanding I suggest to add the deceptive characteristics – BMI, height and weight for different ages and diagnoses in the table together with the information how the food allergy was diagnosed and what symptoms children presented – e.g. diagnosis based on the elimination and provocation diet, based on sIgE results, diagnosed by physician. We could have 10 times higher frequencies of food allergy based only on self-reported data.

The conclusion of the study should be rather that the early onset of persistent atopic dermatitis is more likely related to food allergy, while late onset is less likely. Put in that way it doesn’t suggest causal relationship. It is known that AD could be the risk factor for developing sensitisation to other allergens – because of the damage of the skin barrier, but still at the beginning we could have food allergy as the main initiating trigger for skin lesions – it is difficult to establish if food allergy is a risk or outcome of AD.

Another observation, worthy mention is that majority of AD present in the 1 y of life in transient (8861 in 1y. and 2416 in 3y.) and it is also true for FA+AD (2082 in the 1y. and 904 in the 3y.), after exclusion of possibilities that those are the missing cases.

Authors speculate that chronic inflammation affect the metabolism. Please explain the possible mechanism. More detailed discussion is needed here.

In the abstract and in the discussion there is information about the association of BMI with height – but I can’t see that in the data given. Anyway, is it always true based on the equation?

It will be helpful to add the questionnaire to the manuscript.

6. PLOS authors have the option to publish the peer review history of their article (what does this mean?). If published, this will include your full peer review and any attached files.

Reviewer #1: **Yes: **Mitsuhiro Okamoto

Reviewer #2: No

---

## [Author Response · Author response to Decision Letter 0]

8 Nov 2021

PONE-D-21-29612

Persistent atopic dermatitis leads to both impaired growth and food allergy: JECS Birth Cohort

PLOS ONE

Comments to the Author

1. Is the manuscript technically sound, and do the data support the conclusions?

Reviewer #1: Yes

Reviewer #2: Yes

Response: Noted.

2. Has the statistical analysis been performed appropriately and rigorously?

Reviewer #1: Yes

Reviewer #2: Yes

Response: Noted.

3. Have the authors made all data underlying the findings in their manuscript fully available?

Reviewer #1: Yes

Reviewer #2: No

Response: We have already mentioned the data-sharing policy of our study in our first submission.

Data are unsuitable for public deposition because of ethical restrictions and the legal framework of Japan. The Act on the Protection of Personal Information (Act No.57 of May 30, 2003, amendment on September 9, 2015) prohibits the public deposition of data containing personal information. The Ethical Guidelines for Medical and Health Research Involving Human Subjects enforced by the Japan Ministry of Education, Culture, Sports, Science and Technology and the Ministry of Health, Labour and Welfare also restricts the open sharing of epidemiologic data. All inquiries about access to data should be sent to jecs-en@nies.go.jp. The person responsible for handling inquiries sent to this e-mail address is Dr Shoji F. Nakayama, JECS Programme Office, National Institute for Environmental Studies. URL: https://www.env.go.jp/chemi/ceh/en/index.html. The authors had no special access privileges to data others would not have.

4. Is the manuscript presented in an intelligible fashion and written in standard English?

Reviewer #1: Yes

Reviewer #2: Yes

Response: Noted.

5. Review Comments to the Author

Response: Noted.

Reviewer #1: General Comments:

This paper used a large prospective cohort study to investigate atopic dermatitis and food allergy and body size. The weakness of this paper is that the diagnosis of food allergy is not clear. It is likely that most children did not receive appropriate oral food challenge tests during the follow-up period because the number of children with food allergies changed little between the ages of one, two, and three years. Therefore, I frankly felt that children with persistent atopic dermatitis are more likely to be misdiagnosed with food allergy. In the future, studies with a definitive diagnosis of food allergy will be more valuable than ones with large numbers of subjects. On the other hand, as far as I know, there are no studies of this size or larger, so this study is of great value. Hence, I believe that this study is worthy of publication.

Response: Thank you for your kind comments. The discrepancy in the diagnosis of food allergy has been described in the Discussion section. Our study was not performed in hospitals or clinics, as we evaluated the entire general population with and without food allergy. Furthermore, an oral food challenge test was performed only at a certified allergy department. Therefore, it was not feasible to conduct the oral food challenge test for all general children in our study. Regarding the outcome assessment, various studies in high-impact journals have reported the use of the physician’s diagnosis of food challenge via the caregiver’s report. We believe that our study outcome assessment is acceptable. A systematic review of 66 studies concluded that atopic dermatitis appeared to precede the development of food allergy (Tsakok T, Marrs T, Mohsin M, Baron S, du Toit G, Till S, Flohr C. Does atopic dermatitis cause food allergy? A systematic review. J Allergy Clin Immunol. 2016 Apr;137(4):1071-1078). Therefore, we consider that atopic dermatitis occurs before food allergy.

Specific recommendations for revision minor:1. Reference 18 shows that patients with obesity are more likely to have complications of atopic dermatitis. This does not seem to be relevant to the discussion of this study. For example, I propose to discuss reference "Zhang A, et al. Association of atopic dermatitis with being overweight and obese: a systematic review and metaanalysis. J Am Acad Dermatol. 2015; 72: 606-16.e4." as a foundation. Excluding severe cases, such as those involving sleep disorders, atopic dermatitis was not thought to affect the physique. It is interesting that your study found that growth retardation was present at age 3 years.

Response: Thank you for this informative comment. We have referred to a different reference. We were supposed to discuss the same study (Zhang A, et al. Association of atopic dermatitis with being overweight and obese: a systematic review and metaanalysis. J Am Acad Dermatol. 2015; 72: 606-16.e4) that the reviewer has proposed here. We have updated reference 18; however, we have not revised the description of the discussion.

Reviewer #2: The strength of this study are impressive numbers. It’s also interesting and definite.

We know that chronic conditions could impair the growth and development of children, it is anything new here. What’s interesting, as authors have mentioned in the discussion – in other studies no such impairment was observed in AD – it will be highly welcomed to discuss this in more detailed way - why the differences between current study and previous reports are present – whether it is the age of children, comorbidities or other factors.

The coexistence of atopic dermatitis and food allergy – we don’t know what comes first in infants. The most possible course is that food allergy if the first phenomenon with the presentation of symptoms from the skin – that’s why many infants improve on elimination diet. Another aspect not discussed in the manuscript is how elimination diet could impair the growth.

Response: Thank you for these important comments. Accordingly, we have added an explanation to the Discussion section. To the best of our knowledge, this is the first report on the relationship between infant AD and physical growth among the general Japanese population of preschool children. In addition, we reported the difference between AD phenotypes and physical growth, although prior studies evaluated only the relationship among all ADs, including all phenotypes and physical growth. We believe that these are the interesting points of our study. Although we mentioned “the present study speculates that severe and persistent AD may lead to persistent skin inflammation, producing various cytokines/chemokines from the skin that may, in turn, affect systemic metabolism” in the Discussion section, a search of past studies did not reveal any reports on the detailed mechanism of infant AD and physical growth, including comorbidities and other factors. Unfortunately, we could not add many references to the Discussion section. However, a systematic review of 66 studies has already concluded that atopic dermatitis appears to precede the development of food allergy (Tsakok T, Marrs T, Mohsin M, Baron S, du Toit G, Till S, Flohr C. Does atopic dermatitis cause food allergy? A systematic review. J Allergy Clin Immunol. 2016 Apr;137(4):1071-1078). Therefore, we consider that atopic dermatitis occurs before food allergy. We have added the explanation following the comments on Line 188-192. As for the elimination status, we did not evaluate the details of the elimination diet. However, we believe that food allergy guidelines recommend minimum causal food elimination; thus, we consider that most children underwent only minimum causal food elimination that did not affect physical growth. We have added this information within the discussion of the study limitations on 206-210.

For better understanding I suggest to add the deceptive characteristics – BMI, height and weight for different ages and diagnoses in the table together with the information how the food allergy was diagnosed and what symptoms children presented – e.g. diagnosis based on the elimination and provocation diet, based on sIgE results, diagnosed by physician. We could have 10 times higher frequencies of food allergy based only on self-reported data.

The conclusion of the study should be rather that the early onset of persistent atopic dermatitis is more likely related to food allergy, while late onset is less likely. Put in that way it doesn’t suggest causal relationship. It is known that AD could be the risk factor for developing sensitisation to other allergens – because of the damage of the skin barrier, but still at the beginning we could have food allergy as the main initiating trigger for skin lesions – it is difficult to establish if food allergy is a risk or outcome of AD.

Response: Thank you for the comments. We added information regarding how we defined the outcomes to the table legends. We have also added an explanation in the Discussion section following the comments. 

Line 188-192: “A systematic review of 66 studies concluded that atopic dermatitis appeared to precede the development of food allergy ((Tsakok T, Marrs T, Mohsin M, Baron S, du Toit G, Till S, Flohr C. Does atopic dermatitis cause food allergy? A systematic review. J Allergy Clin Immunol. 2016 Apr;137(4):1071-1078). Therefore, we considered that atopic dermatitis occurs before food allergy. It is also known that AD can be a risk factor for developing IgE sensitization to allergens. We considered that eczema occurred first, sensitization second, and food allergy third based on the systematic review.”

Another observation, worthy mention is that majority of AD present in the 1 y of life in transient (8861 in 1y. and 2416 in 3y.) and it is also true for FA+AD (2082 in the 1y. and 904 in the 3y.), after exclusion of possibilities that those are the missing cases.

Response: Thank you for this comment. We agree with it. Most cases of AD at the age of 1 year were transient. We have added an explanation to the Results section on Line 137-138.

Authors speculate that chronic inflammation affect the metabolism. Please explain the possible mechanism. More detailed discussion is needed here.

Response: Thank you for this comment. We agree with it. Accordingly, we have added an explanation to the Discussion section. 

Lines 195-197: A previous study reported that severe childhood AD led to hypoprotenemia, hyperkaremia, and hyponatremia as leaking through the skin. [22] Homeostasis of the body could be damaged by skin inflammation.

Nomura I, Katsunuma T, Tomikawa M, Shibata A, Kawahara H, Ohya Y, et al. Hypoproteinemia in severe childhood atopic dermatitis: a serious complication. Pediatr Allergy Immunol. 2002;13(4):287-94. Epub 2002/10/23. doi: 10.1034/j.1399-3038.2002.01041.x. PubMed PMID: 12390445.

In the abstract and in the discussion there is information about the association of BMI with height – but I can’t see that in the data given. Anyway, is it always true based on the equation?

Response: We have made mistakes on these points. However, the results presented in the table are correct. We revised the explanation in the abstract along with the table. Also, we have revised the explanation in the Discussion section.

In the abstract: Persistent AD was negatively associated with height at the age of 2 years (estimated coefficient, −0.127; 95% CI, −0.16 to −0.095) and 3 years (−0.177; 95% CI, −0.214 to −0.139)). The same tendency was also observed with weight and BMI.

Lines 175-177: Data obtained from a large-scale, national, birth cohort study in Japan revealed that early-onset and persistent AD negatively affected physical growth and created a risk of low body weight, short height, low BMI, and development of food allergy.

It will be helpful to add the questionnaire to the manuscript.

Response: We understand this point. However, the JECS study group has not completed the preparation of the questionnaire for the public. However, the Ministry of the Environment, Japan is preparing to share the questionnaire via the study website. All can access the questionnaire via the study website. 

6. PLOS authors have the option to publish the peer review history of their article (what does this mean?). If published, this will include your full peer review and any attached files.

Do you want your identity to be public for this peer review? For information about this choice, including consent withdrawal, please see our Privacy Policy.

Reviewer #1: Yes: Mitsuhiro Okamoto

Reviewer #2: No

Response: Noted.

---

## [Editor Report · Decision Letter 1]

10 Nov 2021

Persistent atopic dermatitis leads to both impaired growth and food allergy: JECS Birth Cohort

PONE-D-21-29612R1

Dear Dr. Yamamoto-Hanada,

We’re pleased to inform you that your manuscript has been judged scientifically suitable for publication and will be formally accepted for publication once it meets all outstanding technical requirements.

Kind regards,

Kazumichi Fujioka

Academic Editor

PLOS ONE
---

## [Editor Report · Acceptance letter]

19 Nov 2021

PONE-D-21-29612R1 

Persistent eczema leads to both impaired growth and food allergy: JECS Birth Cohort 

Dear Dr. Yamamoto-Hanada:

I'm pleased to inform you that your manuscript has been deemed suitable for publication in PLOS ONE. Congratulations! Your manuscript is now with our production department. 

Kind regards, 

on behalf of

Dr. Kazumichi Fujioka 

Academic Editor

PLOS ONE